# Shining a light on camouflage evolution: Using genetic algorithms to determine the effects of geometry and lighting on optimal camouflage

George R. A. Hancock[1]*, Innes C. Cuthill[2], Jolyon Troscianko[1]

1 Centre for Ecology & Conservation, University of Exeter, Penryn, United Kingdom, 2 School of Biological Sciences, University of Bristol, Life Sciences Building, Bristol, United Kingdom

* ghancockzoology@gmail.com

## Abstract

Visual camouflage evolution in animals depends on both light's interaction with its surroundings and the limits of what the natural observers of the camouflage can see. Changes in lighting – such as those caused by shifting weather – can quickly alter how animals and their surroundings appear due to the generation of shadows both cast onto objects and from self-shading. The extent and nature of these changes will depend on the three-dimensional (3D) structure of the environment. Despite the apparent effects of lighting on object and background appearance, the enormity of interactions and the diversity of animal camouflage methods and appearances pose challenges to investigating the combined effects of lighting and habitat structure on camouflage. Genetic algorithms and mathematical methods for generating animal patterns provide a potential solution for investigating camouflage's broad feature space by evolving artificial prey under different lighting conditions within different habitats. Here, an online artificial evolution experiment was used to examine the effect of lighting, both direct and diffuse, and habitat geometry on camouflage. Lighting and geometry changed the appearance of the evolved prey, and the predictive power of common measures of camouflage. Crucially, lighting condition systematically altered how contrasting the prey-targets' internal patterning was and interacted with habitat geometry to affect the evolved pattern shapes, colours, and countershading. Our work demonstrates the importance of considering the relative geometry and lighting of an environment when determining the function of animal colouration and the adaptive value of camouflage.

## Introduction

Animal camouflage has long served as an example of evolution by natural selection, as the colours and patterns used by animals reflect the characteristics of their local environment as seen by the visual systems of their observers, be they predators

**Data availability statement:** All relevant data are within the paper and its Supporting Information file.

**Funding:** Natural Environment Research Council (NERC) GW4+ NE/S007504/1 funded GRAH in a CASE partnership with the Game and Wildlife Conservancy Trust, UK. JT was funded by a NERC Independent Research Fellowship NE/P018084/1 and ICC by Biotechnology and Biological Research Council grant BB/S00873X/1.

**Competing interests:** The authors have declared that no competing interests exist.

or prey [1,2]. Effective camouflage, however, does not simply rely on matching the colour and pattern of the local background [2–6]. Recognition, saliency, edge disruption and shape destruction all play a role in camouflage [7–12]. Background matching itself must function within the variable appearances of an animal's background through space and time. Most animals move and so interact with – and are observed against – different surfaces and backgrounds [13,14]. Meanwhile, the range, frequency and pattern of colours in natural environments shift with the seasonality of their biotic (e.g., plant phenology) and abiotic (e.g., surface moisture) components [15,16]. Even if animals and their surrounding surfaces were somehow locked in time, their visual appearance would still vary from changes in lighting conditions caused by weather and the sun's position [17,18]. Weather in particular poses a challenge to camouflage as passing clouds and wind-swept foliage can rapidly change the lighting of scenes [19,20]. This instability of lighting necessitates behavioural, morphological and life history specialisations to maintain effective camouflage, as well as to maximise salience when signalling [4]. In this paper, we combine artificial animal pattern generation, genetic algorithms and mass-participation online games to explore the evolution of camouflage in environments with different lighting conditions.

Firstly, what should camouflage that is specialised to either fixed or variable lighting environments look like, and how does it function? To understand these questions, we must first summarise how lighting changes the visual scene. On a cloudless day, when the habitat is open, the lighting of the environment is intense and directional. Surfaces facing the source of light are illuminated more intensely, while surfaces obstructed by their own or surrounding structures are in shadow. Shadows cast onto an object's surface by itself (self-shadows) and onto surrounding surfaces (cast shadows) can provide cues to their shape and location [21,22]. Obstruction of light does not simply change the luminance contrast; it also changes the ratio of longwave to shortwave light, because objects in shadow are illuminated by the sky, not the sun, and Rayleigh scattering is wavelength-dependent [23,24]. For human trichromat vision, this results in an increase in the blue-yellow contrast of the scene [25], with shaded regions appearing bluer compared to regions in direct sunlight (Endler, 1993). Meanwhile, atmospheric conditions (e.g., cloud/fog) or the sun's elevation (e.g., dusk/dawn) can scatter the light illuminating the scene, resulting in a diffuse lighting environment. Scenes with diffuse lighting still possess depth cues from self-shading, although they areless contrasting (Penacchio et al., 2015; Cuthill et al., 2016), and lack salient cast shadows [7,26]. How these changes influence the detectability of animals is poorly understood. The increased internal contrast and the presence of cast shadows under direct lighting may allow animals to be more easily distinguished from their background while increased visual complexity of the background from shadows and disruption of the animal's shape from received shadows may instead hamper detection.

The disparities between visual scenes under different atmospheric conditions are influenced by the surrounding geometry of the scene. Flat surfaces allow 3-dimensional (3D) objects to cast unobstructed shadows, providing salient cues for their depth and location within the scene [27].Within geometrically complex

environments, light reflects from objects and scatters, becoming more diffuse, before reaching the scene or target object (Endler, 1993). Environments with tall vegetation – relative to the observer or target – therefore have less intense lighting, but can also feature dynamic dappled lighting and shadows from the movement of leaves and branches [17,19,23]. This, in turn, can affect the detectability of objects [19,20]. Even shorter ground cover can influence the conspicuousness of cast shadows and self-shadows. Coarse surfaces, such as vegetation, distort the shape of and interfere with shadows [27,28]. Tall thin branches and grasses have the added effect of creating thin directional shadows on surfaces, changing the uniformity of direction of patterns within the scene (directionality/anisotropy) under direct lighting. These sharp contrasting shadows disappear under diffuse lighting, though geometry will still affect and create soft shadows from differences in depth. Thus, how lighting influences camouflage should depend on the frequency of different atmospheric conditions and the geometry of an animal's habitat across spatial scales relative to its size.

Existing research on how changes in the directionality and intensity of light from weather and geometry influence animal camouflage has largely been observational for all but a few mechanisms. Flat habitats and surfaces are thought to select for flatter postures and body plans in marine (flat-fish), terrestrial (bearded dragons) and trunk-dwelling species (leaf-tailed geckos, moths) [4]. Flattening reduces the conspicuousness of cast and self-shadows and other pictorial depth cues. In the inverse setting, shadows of closed 3D complex habitats are thought to select for darker and or more complex and contrasting luminance patterns based on phylogenetic and ecological comparisons of felid coat patterns [29–33]. Experimentation on pattern adaptation to light environments has predominantly focused on countershading and edge-enhancement camouflage. For example, comparisons of the patterns of ungulates have shown that the intensities of their countershading gradients (the downward increase in luminance from their dorsal to ventral surface) are higher in species that occupy open environments at lower latitudes, where the intensity and directionality of the lighting environment are expected to be greater and more consistent [34]. These observations are supported by experiments showing that countershaded models survived for longer if they had countershading gradients optimised to their self-shading gradient [18]. While edge enhancement patterns, which pictorially mimic shadows that form at the edges of objects from direct lighting [35], are harder to find under direct rather than diffuse illumination [36].

How shadows should influence the orientation and colour of animal patterns is less clear. Cuttlefish can adaptively adjust the size and shape of their pattern to match the geometric and pictorial properties of their surrounding and produce shadow-mimicking patterns within directly illuminated 3D scenes [37,38]. While a number of other invertebrates, fish, amphibians and reptiles can adjust their luminance, albeit at a slower rate, most terrestrial animals cannot perform rapid adaptive colouration and must make trade-offs or specialise to their environment [39]. The reduced stability of blue-yellow compared to green-red in neural opponent processing under variable lighting is thought to be one factor that drives the use of red colouration in animal signalling and for the detection of camouflaged predators by primates [40–42]. But does the stability of information change the intensity of selection for background matching within these channels? Do long shadows from 3D complex habitats select for more contrasting and directional patterns and how might changes in light paths from local geometry affect camouflage strategies such as countershading, edge enhancement and glossiness [18,29,36,43]? How does the directionality of light interact with features of an animal's texture that affect camouflage, e.g., Glossiness [44]? Moreover, how camouflage colours and patterns should be generalised to changes in background scene statistics from the weather is unknown. To simplify this host of potential phenotypic interactions with the directionality of light and habitat geometry, we have provided a table of predictions (Fig 1).

The vast array of interactions between lighting and geometry present fundamental challenges to examining their effects on camouflage [45]. The scene's 3D geometry must be measured at multiple spatial scales and its visual appearance imaged under different weather conditions. Simultaneously, a large gamut of both intersecting and non-overlapping camouflage strategies must be compared. Here we aimed to provide a cohesive investigation of the broad effects of lighting and geometry on optimal animal camouflage colouration and patterns. We opted for a heuristic search approach, using an online citizen science game derived from the CamoEvo toolbox [46], which we used to disentangle camouflage

## Summary of Predictions

| Phenotype | Predictions | | | Justification for Predictions |
|---|---|---|---|---|
| Low -> High | Direct Lighting | 3D Variation | Light * 3D Interaction | |
| CIE L* Luminance | ☀ ↓ | ⏷ ↓ | ☀ ⏷ ↓ (large) | Shadows formed under direct lighting and from occlusion within 3D complex habitat should result in darker targets (Cheng et al, 2018). |
| CIE b*, Blue-Yellow | ☀ ↓ | ⏷ ↑ | ☀ ⏷ ↓ (large) | As with the above, short wave shifted shadows should result in bluer targets, but increased 3D variation correlates with green-yellow vegetation (Steverding and Troscianko, 2004). |
| CIE a*, Green-Red | ☀ — | ⏷ ↓ | — | Green-red colouration is more stable to changes in lighting, but it also correlates with the 3D environment with more vegetated habitats being greener (Arenas et al, 2014; Lovell et al, 2005). |
| Pattern Contrast | ☀ ↑ | ⏷ ↑ | ☀ ⏷ ↑ (large) | Luminance contrast is greater within more geometrically complex environments and especially under direct lighting (Allen et al, 2011). |
| Pattern Directionality | ☀ ↑ | ⏷ ↑ | ☀ ⏷ ↑ (large) | The patterns of felids suggest that more geometrically complex habitats, e.g. closed, and shadows should promote more directional patterns (Allen et al, 2011). |
| Countershading | ☀ ↑ | ⏷ — | ☀ ⏷ ↑ (small) | Countershading should be proportional to the self-shading, resulting in stronger countershading within open habitats under direct lighting (Cuthill et al, 2016; Penacchio et al, 2018). |
| Edge Enhancement | ☀ ↑ | ⏷ ↑ | ☀ ⏷ ↑ (large) | Direct lighting increases background boundary contrast due to shadows and specular highlights and so targets should have more edge enhancement under direct lighting (Egan et al, 2016). |
| Glossiness | — | — | — | Glossiness should just decline over time as most backgrounds lack glossy features and specular highlights can give away an object's shape and position (Thomas et al, 2023). |

**Fig 1. A summary of the predicted effects of lighting condition, habitat 3D variation and the interaction between direct lighting and 3D variation on the appearance of all animal patterns.** Upwards arrows indicate a positive effect on the phenotypic feature, e.g., increased luminance or directionality, while downwards arrows indicate a negative effect; a horizontal line indicates no predicted effect. Larger arrows indicate that increased variation increases the effect size of direct lighting, while smaller arrows indicate a reduction in effect size. For each prediction, the justification and relevant references are provided. CIE L*, a* and b* refer to the mean values of the opponent colour channels used by the CIELAB perceptual colour space [50]. These channels are L* (luminance, low=dark, high=light), a* (red-green, low=green, high=red) and b* (blue-yellow, low=blue, high=yellow).

optimisation within habitats of varying 3D and spatio-chromatic variability and under direct and diffuse lighting conditions. CamoEvo combines genetic algorithms (GAs), Gray-Scott reaction-diffusion patterns (aka Turing patterns), and human visual search experiments to compare the effects of background features on camouflage optimisation [29,45,47]. Camo-Evo and similar natural selection-inspired mechanisms, such as other GAs and generative adversarial networks, have proved successful in creating functional animal-like camouflage patterns against different experimental backgrounds [47–49]. By using a GA to compare camouflage optimisation between lighting conditions, we can determine whether and how lighting influences the pattern shape, colours and specular reflection of targets, while also validating the circumstances where characteristics known to be influenced by lighting, such as countershading and edge enhancement, are under selection [18,36]. A full table of predictions is provided below.

## Materials and methods

### (a) Visual scenes

For our backgrounds, we used a set of 28 temperate habitats photographed across the UK as part of a previous study on the effects of lighting and geometry on the patterns of visual scenes and objects [27]. These backgrounds included grassland, heathland, woodland and estuarine habitats. All photographs were taken from a height of 1.2 m with an ASUS Zenfone (zenfone AR ZS571KL, asustek Computer Inc., Taipei, Taiwan) and calibrated with the MICA image analysis toolbox [51]. For each habitat, the same 24 scenes were photographed under direct and diffuse lighting conditions. Direct lighting conditions were days with <10% cloud and no less than 2 h from sunrise or sunset. The diffuse lighting photographs of the same background were taken under a 1.5 m³ photography tent placed over the scene (NEEWER., Bee Block Inc, 15 Cotters Lane, East Brunswick, NJ 08816, USA), simulating the isotropic lighting conditions of cloudy days (Fig 2).

Within each scene, a 3D-printed 30 mm diameter button-shaped target painted grey with a matt 8% reflectance paint was placed at the centre. This target was used to normalise each scene using the MICA toolbox, as well as for rendering the evolved pattern. Additionally, for each lighting condition, a repeat photograph was taken with a black unpainted target soaked with acetone to give it a 'glossy' texture. 3D scans of the scenes were taken from the same camera position using the phone's built-in Tango-enabled 3D scanner (Tango being Google's 3D mapping technology) [52]. All scenes were limited to having some unobstructed lighting present (e.g., dappling in woodland), though the target itself could be placed within shadow or direct light within the scene. Habitat 3D variation was measured as the standard deviation of height across the scene averaged across all 24 scenes. Backgrounds with higher 3D variation featured vegetation and other elevated surfaces, e.g., large stones.

### (b) Pattern creation

To create our evolving targets, we used a modified version of CamoEvo's pattern generator [46]. CamoEvo uses Gray-Scott reaction-diffusion generated markings, CIELAB colours [50] and additional image filters to provide edge enhancement and countershading, creating biologically relevant patterns (Fig 2) These phenotypic traits of the target patterns were all controlled by 34 decimal genes on a haploid chromosome. These genes in turn were passed to CamoEvo's GA to mutate, recombine and select individuals. A spherical pattern deformation function was applied to the patterns for our experiment to create realistic 3D wrapping, and two additional genes/traits were added for controlling glossiness of the markings and the glossiness of the background pattern. For full details of this system, see the Supplementary Materials (S1_File) and our CamoEvo paper [46].

### (c) Stimuli

To show our evolving patterns under the lighting conditions of the scene, the patterns were rendered onto the calibrated grey target by first subtracting the target by its known reflectance and then multiplying it by the RGB values of the CamoEvo

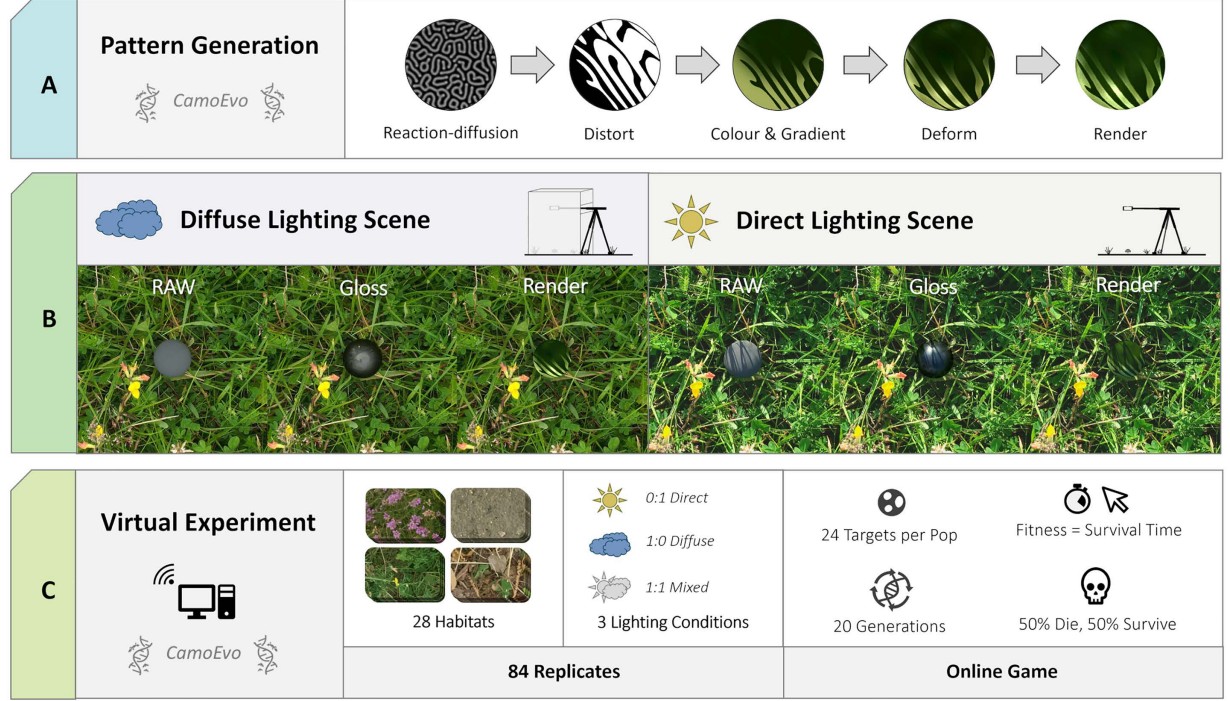

**Fig 2. Schematic for experimental design using a meadow habitat as a demo. (A)** Prey pattern generation using CamoEvo patterns, spherical deformation and rendering. **(B)** Calibrated raw, gloss and rendered targets under diffuse (left) and direct (right) lighting conditions. **(C)** Virtual online predation experiment using 28 different habitats and 3 lighting conditions. Each treatment evolved for 20 generations with 24 targets per population. The selection of targets was based on a hybrid of the time taken to click on and the time taken to respond to the target (survival time). The target pattern used was a sample pattern shown from the last generation (gen=20) of the direct lighting-only evolved condition for the chalk wildflower meadow habitat.

pattern. To change the target from matt to glossy, the image of the glossy target was combined with the patterned target using image addition. To provide a range of glossiness values, the image of the glossy target was multiplied by a genetically controlled alpha value (see S1_File for more details). Background images with rendered patterns were then cropped with the target position randomly offset from the centre to ensure the target wasn't always in the same position (S1_File). Each cropped image was 1904 × 1488 px, and the target button had a diameter of 80 px. The target positions were never less than 2x their diameter from the centre or border of the image. These no-go zones were to prevent biases to fitness within the visual search experiment from being too close to the periphery of the observer's vision or immediately fixated at the centre of the screen.

### (d) Online experiment design

For our game the 28 habitats were used for three different lighting treatments: two specialist lighting treatments, where all background images were from direct (DIRECT) or diffuse (DIFFUSE), and a mixed treatment, where half of the backgrounds were under diffuse and direct lighting (MIXED). This gave a total of 84 replicates, 28 per lighting treatment. For each replicate a starting population of 24 prey targets were generated using CamoEvo [46]. The genes of the starting population were set with a uniform distribution from 0–1 across all individuals to prevent initial clustering of genotypes within populations. These targets were rendered onto the target button and backgrounds to create the stimuli above. An online computer-based search game (https://CamoEvo.visual-ecology.com/) was created by modifying JavaScript used for Troscianko et al., 2020 's crab camouflage experiment. This experimental set-up used humans as a surrogate predator to drive selection for the artificial prey for 20 generations (0–20).

Whenever a player played the game, they were randomly assigned one of the 84 populations and were tasked with finding and capturing the targets, as fast as they could. Each population was shown to one participant and once it had been played it could not be played again until all other populations at the same concurrent generation had been played to prevent one population from being ahead of the others. All 24 stimuli and their respective targets were displayed in a random order. At the start of the game, instructions were provided on what the game was for and what the players needed to do to play the game. As participation was anonymous, participants were allowed to play the game multiple times.

As a control for starting viewpoint, participants were tasked with moving their cursor to the centre of the screen before initialising each slide or clicking in the centre for touchscreens. A ring at the centre of the screen marked where the participant needed to move their cursor, with the instruction to not move their cursor until they had spotted the target. To capture the targets, participants needed to click on them within a time window of 15 s. The capture time was coded as the time taken to click on the target, while response time was encoded as the time taken to leave the centre circle. Fitness was encoded as the response time unless a delay of 600 ms was exceeded or the response time was faster than the stimulus display, in which case the capture time was used in place of response time. This measure was used as a proxy for survival time, following the same protocol successfully used for the CamoEvo toolbox [46]. Response and capture time were equal for touchscreen users. Using response time and a centre fixation point minimises fitness noise caused by cursor travel time [53].

Before any member of the target population was shown, a demo target was displayed. This demo consisted of a randomly generated pattern for the first generation (Gen 0) and then a deceased pattern from the previous generation for all subsequent generations. The purpose of this demo-target was to allow the participant to create an initial search image for the targets, which might otherwise make the first target harder to find and so have a disproportionately high fitness [54]. Once a population had been completed, it was removed from the server, patterns were ranked by fitness, and CamoEvo's GA was used to create the next generation. For each round of selection, the 12 lowest (50%) ranking members of the population died, and new hybrid offspring were created by random pairing and recombination of the survivors from the previous generations. The new population for the next generation was then uploaded to the server to be played. This process was repeated until all populations had undergone 20 generations of selection and evolution. As an additional control for noise, the top 4 highest ranking targets from the generation prior to selection were given a lifeline against deletion, again following the same method used for CamoEvo [46].

This project has been reviewed by the CEC Cornwall Ethics Committee at the University of Exeter (R-489629). All participants consented to using their data for the experiment following instructions at the start of the game. Participants were free to exit the game and no identifying data was collected.

## (e) Colour analyses

For each scene, the 3D scans and direct and diffuse lighting images were rescaled to the minimum number of pixels per mm of all target images and converted to human CIELAB colour space [50]. For each channel (L*, a* and b*), the contrast (standard deviation) at 6 different spatial scales (spatial frequencies relative to target diameter, [1/64, 1/32, 1/16, 1/4, 1/1, 1/2, 1/1] x) and 4 different orientations [0, 45, 90, 135] were measured with Gabor filters with custom plugins for ImageJ v 1.53 [55–57]. These measures were taken for the target pattern before rendering (target-skin), the target pattern post rendering (rendered-target), the local surround (circle radius x2.5 target radius) and the global surround (circle radius x15 target radius). The 'directionality' of patterns was measured for each spatial scale by dividing the orientation with the maximum contrast by the mean contrast. Likewise, the 'verticalness' of patterning was calculated by subtracting the contrast filtered at 90 degrees from the contrast at 0 degrees, such that negative values indicated horizontal patterns and positive values vertical patterns. For the background, we also measured the 3D variability as the standard deviation of depth for the global surround. As a measure of habitat spatial variation in luminance, green-red, blue-yellow and depth, we measured the mean contrast of each channel across all spatial scales, $L_{variation}$, $a_{variation}$, $b_{variation}$, $3D_{variation}$, respectively.

To measure camouflage, we used metrics previously used for computer visual search and field experiments [6,53,58]. All metrics used have been shown to be able to predict the conspicuousness of prey. Luminance match was measured as the absolute difference between the mean of the L* channel of the local background and the rendered-target (luminance$_{difference}$), while colour match was the Euclidean distance in the green-red (a*) and blue-yellow (b*) plane (colour$_{difference}$) [54]. Pattern difference was measured as the sum of absolute difference in contrast at each spatial scale for the luminance channel (pattern$_{difference}$) [6]. Target edge disruption was measured using gabrat of the target's surrounding in the luminance channel, with the default sigma of 3, against its original background [53]. Gabrat works by quantifying the ratio of "false" edges to "coherent" edges around the object's outline and has been shown to predict search time for camouflaged disruptive prey by humans.

Novel pattern measures were also used. To measure countershading and self-shading, the linear gradient from the top to the bottom of the target was measured for the CIELAB channels. Target gradients were measured for the target-skins, averaged direct lighting raw-targets for each habitat (see S1_File) and the targets rendered onto the averaged raw-target of its corresponding habitat. A positive gradient indicated that the top of the target was lighter than the bottom. Target glossiness was measured by calculating the difference in luminance value (mean and Std Dev) with and without the gloss genes active for each rendered target, as well as with the maximum gloss. Maximum glossiness was measured as the luminance and pattern of the targets could also affect glossiness; darker targets appeared glossier than lighter targets. Edge enhancement was measured as the difference in luminance pattern contrast at 4 spatial frequencies [1/64, 1/32, 1/16, 1/8] between the target with and without (no edge enhancement) the edge enhancement genes active.

## (f) Statistical analyses

Statistics were performed with R, version 4.3.0 [59]. All metrics for target phenotype, camouflage and background structure were continuous and were re-scaled to a mean = 0 and StdDev = 1, lighting treatment was encoded as a categorical variable with three-factor levels (DIRECT, DIFFUSE and MIXED). To test the effects of lighting and habitat 3D variation (3D$_{variation}$) on our proxy for fitness (survival time) we used linear mixed models with the lme4 package [60]. The rescaled log survival time was given as our predictor variable with generation number (N-Gen), lighting and 3D$_{variation}$ as our fixed effects with the interaction between the three variables. We also compared the effect of lighting and 3D$_{variation}$ on the predictive power of our camouflage metrics (luminance$_{difference}$, colour$_{difference}$, pattern$_{difference}$, gabrat$_{difference}$) for fitness, using each camouflage metric, instead of N-Gen, as the fixed effect. For all analyses, the habitat and population were used as random effects to account for changes caused by lighting between populations within the same habitat and individual population effects. Using lme4 nomenclature, the models were as follows: lmm(log(survivaltime) ~ camouflagemetric * Light * 3D + (1|Habitat) + (1|Population) …). The use of * indicates interaction between fixed effects.

To compare the effects of lighting on the evolution of phenotypes predicted to be affected by direct lighting and its interaction with geometry (i.e., pattern contrast, countershading, pattern directionality, pattern orientation, edge enhancement and glossiness), we first compiled the pattern metrics of the target-skin (contrast, directionality, verticality and gradient in the L*, a* and b* channels). These phenotype metrics were then used in a principal component analysis to account for multicollinearity between variables and whether treatment explained any of the principal components. We then used a similar model structure to the above to compare how lighting and geometry affected the evolution of principal components and our phenotype metrics. The phenotype metrics and principal components were used as the response variables, while N-Gen, lighting and 3D$_{variation}$ were used as the fixed effects with population and habitat as random factors. Using lme4 nomenclature, the models were as follows: lmm (phenotypemetric ~ N-Gen * Light * 3D + (1|Habitat) + (1|Population) …). To ensure model assumptions were met, all residuals were checked for homogeneity of variance and homoscedasticity. Square root and log transformations were used as required to meet data distribution criteria.

## Results

For brevity, unless stated otherwise, all effects are significant at p < 0.001, the conservative threshold being necessitated by the large number of tests conducted. Full model output tables are provided in the Supplementary Materials (S2_File).

## 1. Camouflage: fitness, match and edge disruption

We received the total of 1,764 individual players necessary for the experiment over 12 months, totalling 42,336 target presentations of which only 312 were timeouts. To assess whether targets become more camouflaged, we measured the change in survival time (i.e., our fitness proxy) as a function of generation number (Fig 3). On average, fitness improved with generation number (ei.e. excluding lighting treatment and 3D complexity as predictors; N-Gen, $\beta = 0.035$, $t_{42250} = 40.37$) with a starting median survival time of 649 (IQR = 667.5) milliseconds (ms) and an ending median of 1026 (IQR = 1130.25) ms. Of the 84 treatment populations, 65 improved, and the remaining 19 (DIRECT = 4, DIFFUSE = 6, MIXED = 9) either did not significantly increase in fitness (N = 6) or decreased in fitness with generation number (N = 12). The evolved fitness was highest for the population evolved against the gravel background under mixed lighting, with a median survival time of 1321 (IQR ± 3119) ms. Populations where the generation number lowered survival time were excluded from subsequent analyses, with the exception of the habitat analysis below.

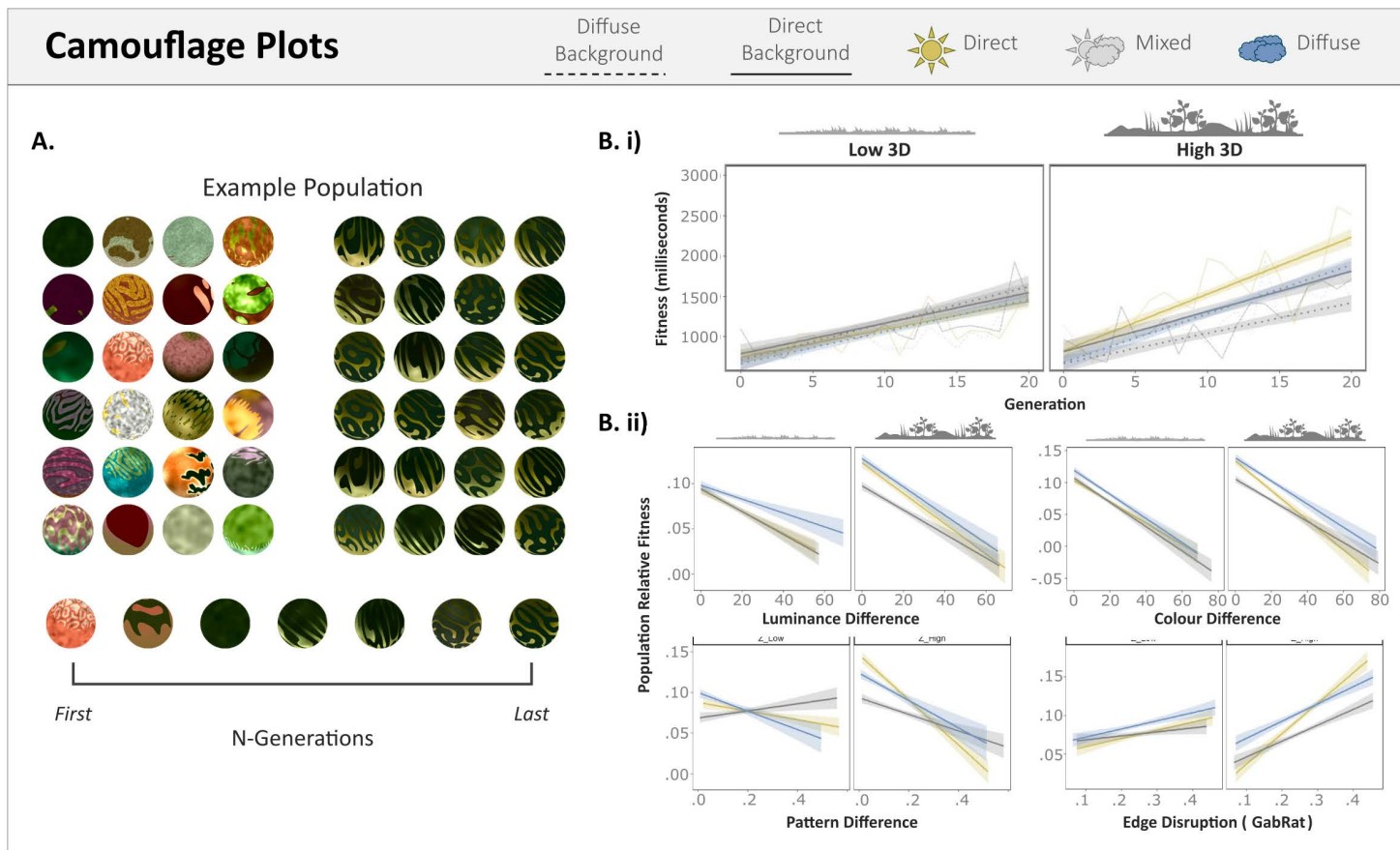

**Fig 3. Evolution of Camouflage. A.** Example change in phenotypic appearance for a population that improved in fitness (survival time). B.i) Average capture time increased over successive generations. Line colour is used to indicate the lighting treatments. These are direct (yellow), diffuse (blue), or mixed (grey, 50:50 direct and diffuse lighting). Dashed lines are used to indicate when the background is diffuse. 3D complexity is split into low (below the mean) and high (above the mean). B.ii) Each pane shows the log predictive value (slope) of different common camouflage metrics for the survival time of the camouflaged targets. These are luminance difference (absolute CIE L*difference), colour difference (Euclidean CIE a* and b* difference), pattern difference (sum CIE L* contrast difference across scales) and edge disruption (gabrat L*). Lower values for the difference metrics (luminance, colour, and pattern) indicate a greater match to the background. As a result, they should have a negative slope for fitness. For gabrat, higher values indicate greater edge disruption and so should instead have a positive correlation with fitness.

We tested the effects of spatio-chromatic complexity in the L*, a* and b* channels as well as 3D variation on the slope of fitness with generation numbers to determine whether properties of the background influenced the evolution of camouflage. In doing so, we found that the slope of fitness was significantly higher for populations on more variable backgrounds for all 4-background metrics (L, a, b and 3D variation, '*' removed from models to avoid confusion with interaction symbology). To account for instances where survival time failed to improve, a binary variable for whether the population had a negative slope for survival time with generation numbers was used as a random factor. The greatest effect size of any habitat metric on target fitness and the level of increase in fitness with generation number was observed for high variance in the combined L* and a* channels ($La_{variation}$, $\beta = 0.034$, $t_{42190} = 35.10$; $La_{variation}$ * N-Gen, $\beta = 0.0127$, $t_{42330} = 13.5$). The highest fitness effect of any individual background metric was from background L variance ($L_{variation}$, $\beta = 0.0273$, $t_{42330} = 15.00$; $L_{variation}$ * N-Gen, $\beta = 0.0009$, $t_{42330} = 5.767$), while simultaneously having the lowest effect on fitness with generation number. Whereas, for improvement in fitness with generation number, it was background B variance which had the largest effect ($B_{variation}$, $\beta = 0.0020$, $t_{42330} = 11.35$, $p = 0.276$ | B * N-Gen, $\beta = 0.0022$, $t_{42330} = 14.24$). Most populations that failed to significantly improve in survival time evolved against low background variation habitats.

As direct lighting increases background luminance and blue-yellow variation, both of which increased fitness (reduced detectability), it is unsurprising that direct-lighting-only treatments evolved targets with a greater fitness than in mixed and diffuse treatments (model using lighting and 3D as factors, as described in methods; N-Gen, $\beta = 0.050$, $t_{36210} = 31.08$; N-Gen * MIXED, $\beta = -0.009$, $t_{36210} = -4.17$; N-Gen * DIFFUSE, $\beta = -0.004$, $t_{36210} = -1.90$). As 3D variation correlates with spatio-chromatic variation, targets evolved to be significantly harder to find when their habitat's 3D variation was higher (N-Gen * $3D_{variation}$, $\beta = 0.009$, $t_{36210} = 5.90$), but had lower survival times for diffuse and mixed lighting treatments (DIFFUSE * $3D_{variation}$, $\beta = -0.025$, $t_{2,41} = -2.43$, $p = 0.020$; MIXED * $3D_{variation}$, $\beta = -0.022$, $t_{2,41} = -2.10$, $p = 0.042$). For populations evolved under mixed lighting, targets had lower survival times when against diffuse lighting backgrounds (effect of background type on survival times for MIXED treatments; $diffuse_{background}$, $\beta = -0.014$, $t_{12070} = -4.33$), especially when habitat 3D variation was high ($diffuse_{background}$ * $3D_{variation}$, $\beta = -0.011$, $t_{12070} = -3.39$).

Luminance difference, colour difference, pattern difference and edge disruption were all significant predictors of survival time. Colour difference ($\beta = -0.043$, $t_{35500} = -23.53$) was the best predictor of survival time across habitats, followed by edge disruption ($\beta = 0.040$, $t_{33860} = 19.93$), pattern difference ($\beta = -0.036$, $t_{34420} = -18.66$) and lastly luminance difference ($\beta = -0.036$, $t_{35780} = -20.21$). All camouflage metrics had significantly steeper slopes for 3D complex habitats. The effect size of camouflage metrics was impacted by lighting treatment. Diffuse only evolved targets had shallower slopes for luminance match ($luminance_{difference}$ * Diffuse, $\beta = 0.007$, $t_{35910} = 2.75$, $p = 0.006$), colour match ($colour_{difference}$ * Diffuse, $\beta = 0.005$, $t_{35780} = 2.19$, $p = 0.028$), and edge disruption (gabrat * Diffuse, $\beta = -0.014$, $t_{34050} = -5.17$). Whereas mixed lighting populations had a shallower slope for pattern match ($pattern_{difference}$ * MIXED, $\beta = 0.023$, $t_{34240} = 8.58$) and edge disruption (gabrat * MIXED, $\beta = -0.009$, $t_{31400} = -3.32$, $p = 0.001$). For mixed lighting evolved targets, colour and pattern difference had an even lower effect on survival time when 3D complexity was high ($colour_{difference}$ * MIXED * $3D_{variation}$, $\beta = 0.008$, $t_{35910} = 3.12$, $p = 0.002$; $pattern_{difference}$ * MIXED * $3d_{variation}$, $\beta = 0.011$, $t_{35480} = 3.70$), where the difference in colour and pattern between diffuse and direct lighting was greater.

## 2. Colour: mean and contrast

Following the observed decrease in absolute colour difference between targets and their background, the colour space of targets in the final populations reflected the colour space of their habitat and lighting treatment (Fig 4). Notably, within 3D complex habitats, targets evolved to be, and were fitter when, darker than their local background, but less so for DIFFUSE populations (see S1_File). The principal component analysis of the target-skin features (Contrast, directionality and verticality in the L*, a* and b* channels) found PC1 (15.66% variance explained) to consist predominantly of pattern contrast in b*, L*, then a* greater than or equal to 1/4x the target's spatial frequency and a decrease in verticality at those same scales, i.e., increase in horizontal contrast. PC2 (9.76% var) was predominantly colour verticality at intermediate

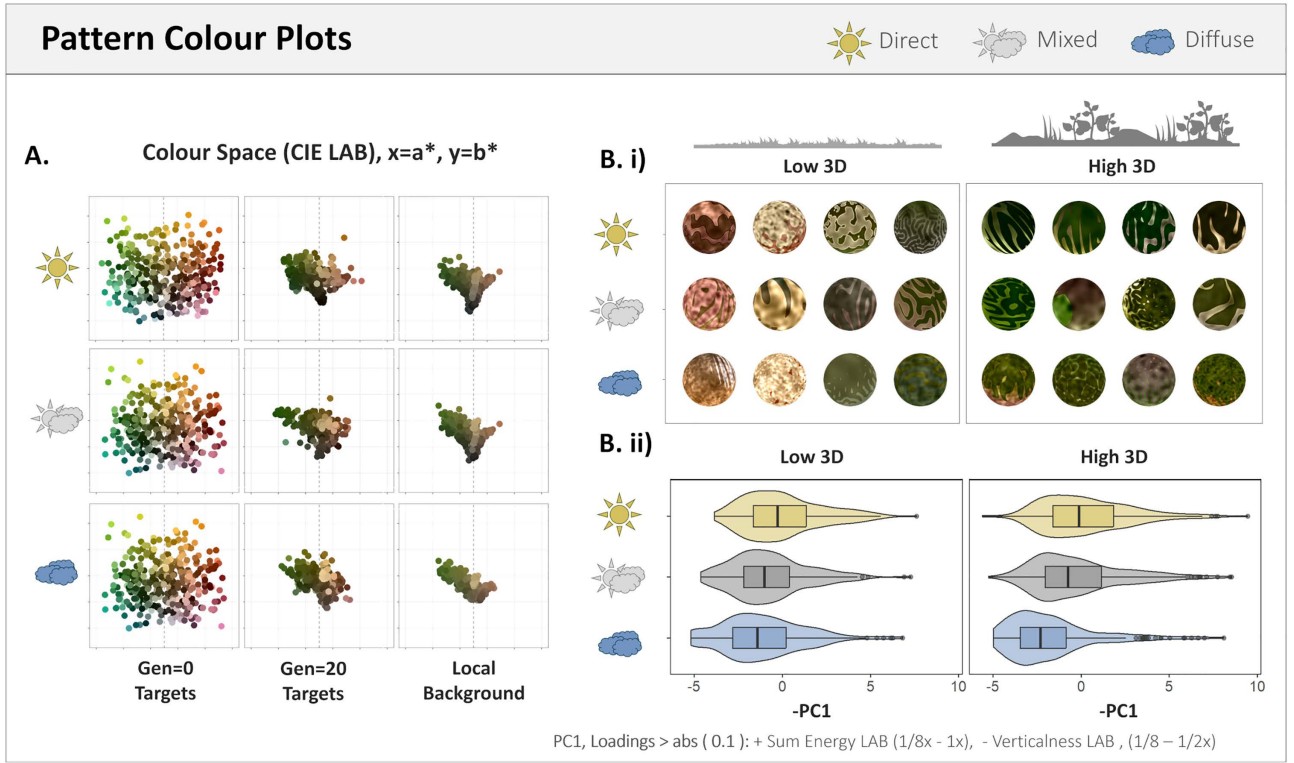

**Fig 4. Change in colour space and colour patterns.** A.i) Populations become increasingly similar to the mean colours of their local backgrounds (2.5x target radius) starting from a dispersed colour space. B.i) Example evolved targets from populations in low and high 3D variation habitats selected for under the different lighting treatments. B.ii) Hybrid violin & boxplots of PC1 for target patterns, where PC1 is predominantly large-scale luminance and colour contrast, and large horizontal (- vertical) patterns. Habitat 3D variation is split by the quantiles for the habitats' mean depth variation across spatial scales.

scales, PC3 was higher green-red (a*) contrast across spatial scales (7.16% var), and PC4 was higher luminance verticality and directionality at small spatial scales (6.77% var). PC1 decreased with generation numbers (N-Gen, $\beta = -0.042$, $t_{36210} = -4.97$), with PC1 being higher for DIRECT populations (DIFFUSE * N-Gen, $\beta = -0.201$, $t_{36210} = -16.63$; MIXED * N-Gen, $\beta = -0.096$, $t_{36210} = -8.04$) and both DIRECT and MIXED populations against more 3D complex habitats but not diffuse (3D * N-Gen, $\beta = 0.026$, $t_{36210} = 3.04$, $p = 0.002$; MIXED * 3D * N-Gen, $\beta = 0.015$, $t_{36210} = 1.24$, $p = 0.214$; DIFFUSE * 3D * N-Gen, $\beta = -0.072$, $t_{36210} = -5.99$). As the pattern contrast and PC1 increases were for the un-rendered targets, differences in target luminance and blue-yellow contrast observed between lighting conditions was not from the self or received shadows on the target and were entirely driven by pattern phenotype evolution.

### 3. Pattern: gradient and shape

Targets in direct lighting populations evolved larger-scale horizontal patterns, as indicated in PCA (above). Countershading requires a dark-to-light dorso-ventral gradient that flattens the internal contrast from self-shading [61]. This gradient should, in turn, be proportional to the intensity of the raw luminance gradient from self-shading, with a steeper gradient required for flatter, more open habitats where the targets have a more intense self-shading gradient. As expected, target populations in direct-lighting-only treatments evolved a more positive gradient (i.e., stronger countershading) (N-Gen, $\beta = 0.022$, $t_{36210} = 20.99$) than those in mixed lighting (N-Gen * MIXED, $\beta = -0.008$, $t_{36210} = -5.36$), and particularly compared to those under diffuse only lighting (N-Gen * DIFFUSE, $\beta = -0.018$, $t_{36210} = -12.00$) when rendered onto the raw direct

lighting target of the corresponding habitat (Fig 5). So, direct lighting targets evolved to be more effective at countershading with mixed lighting as an intermediate. However, the gradient increase was steeper for more 3D variable habitats ($3D_{variation}$ * N-Gen, $\beta = 0.011$, $t_{36210} = 10.11$; $3D_{variation}$ * DIFFUSE * N-Gen, $\beta = -0.008$, $t_{36210} = -5.18$), i.e., when the raw target's luminance gradient was lower rather than higher [18,34]. This reduction of gradient corresponded with the target-skin having a positive luminance gradient and a darker colouration. Although differences in background and target luminance alone could not explain the reduced gradient for the targets within flatter habitats. Comparison of countershading when splitting the habitats by their dominant substrate (bare, gravel, leaf litter, grass and misc-vegetation) found countershading did not affect survival time for gravel habitats and had a lower effect for closed-leaflitter habitats (Fig 5), Table 1). So, lighting environment and geometry both affected and interacted to affect the adaptive value of countershading.

Compared with diffuse lighting treatments, targets under direct and mixed lighting evolved greater pattern contrasts than diffuse at larger spatial scales, especially when 3D complexity was high, but less so for mixed populations (Fig 5), Table 2). At the scale where the marking genes for the target patterns best predicted pattern contrast for the unrendered target (1/32), targets became more directional in their patterning for direct and mixed lighting (N-Gen,

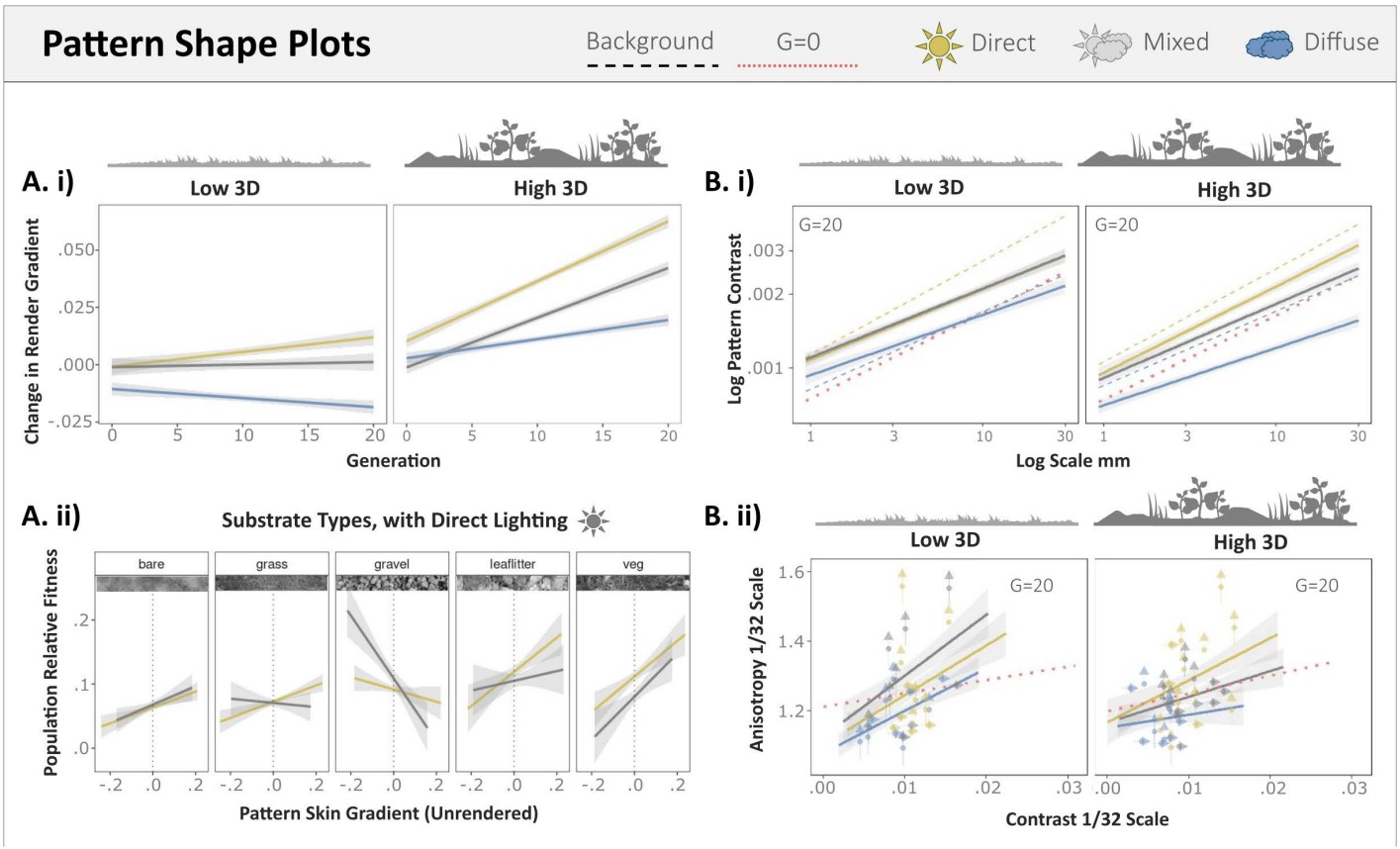

**Fig 5. Phenotypic change in countershading and pattern shape.** A.i) Targets under direct and mixed lighting became increasingly positive in their luminance gradient. A.ii) Having a more positive render target gradient increased survival time for all substrates under both mixed and direct lighting, except for gravel under the direct lighting condition. B.i) Targets evolved under direct and mixed lighting conditions evolve a greater pattern contrast at larger spatial scales and across all scales when background 3D complexity is high. The red dotted line shows the targets in generation zero, dashed lines show the pattern energy of the diffuse and direct backgrounds. B.ii) Under direct lighting contrasting patterns are more directional and orient vertically against 3D complex backgrounds, arrows indicate whether the population average orientation is vertical or horizontal. Habitat 3D variation is split by the quantiles for the habitats' mean depth variation across spatial scales.

**Table 1. Linear mixed model output effects of dominant substrate on the relationship between countershading gradient of the target-skin (CS) and substrate type on target survival time. Positive estimates and t values are shown in blue while negative values are shown in red. Model formula: Lmm (log(Survival Time) ~ CS_Gradient:dominantsubstrate+(1|Treatment)+(1|Habitat)).**

| | Predictor | B | SE | DF | T value | P value |
|---|---|---|---|---|---|---|
| *Fitness* | Intercept | 0.462 | 0.02 | 25 | 30.54 | <0.001 |
| | CS_Gradient:bare | 0.275 | 0.05 | 19120 | 5.46 | <0.001 |
| | CS_Gradient:grass | 0.367 | 0.05 | 18930 | 7.51 | <0.001 |
| | CS_Gradient:gravel | −0.122 | 0.06 | 19140 | −1.92 | 0.054 |
| | CS_Gradient:leaflitter | 0.174 | 0.06 | 19140 | 2.79 | 0.005 |
| | CS_Gradient:veg | 0.345 | 0.06 | 19100 | 5.73 | <0.001 |

$\beta = 0.023$, $t_{36210} = 2.81$, $p = 0.005$; N-Gen * MIXED, $\beta = -0.004$, $t_{36210} = -0.30$, $p = 0.762$) but less directional for diffuse (N-Gen * DIFFUSE, $\beta = -0.126$, $t_{36210} = -10.46$). Directionality further increased with number of generations when 3D complexity was high, but only for direct lighting populations (N-Gen * 3D, $\beta = 0.095$, $t_{36210} = 11.10$) and otherwise declined for mixed and diffuse populations (N-Gen * 3D * MIXED, $\beta = -0.176$, $t_{36210} = -14.35$; N-Gen * 3D * DIFFUSE, $\beta = -0.126$, $t_{36210} = -10.26$), supporting the hypothesis that direct lighting selects for more directional patterns then diffuse lighting. Patterns also became increasingly vertical for direct (N-Gen, $\beta = 0.043_{36210}$, $t = 5.27$) compared to mixed and diffuse (N-Gen * MIXED, $\beta = -0.031$, $t_{36210} = -2.58$; N-Gen * DIFFUSE, $\beta = -0.073$, $t_{36210} = -5.94$). As with directionality, this observed increase in verticality was enhanced for 3D complex habitats for direct but not diffuse or mixed lighting populations (N-Gen * 3D$_{variation}$, $\beta = 0.095$, $t_{36210} = 11.10$; N-Gen * MIXED * 3D$_{variation}$, $\beta = -0.176$, $t_{36210} = -14.35$; N-Gen * DIFFUSE * 3D$_{variation}$, $\beta = -0.126$, $t_{36210} = -10.26$).

### 4. Additional Features: Edge enhancement and glossiness

The positive shift in luminance contrast from edge enhancement increased with generation number(N-Gen, $\beta = 0.043$, $t_{36210} = 5.31$) but less so for diffuse lighting targets (DIFFUSE * N-Gen, $\beta = -0.030$, $t_{36210} = -2.54$, $p = 0.011$). Increased 3D complexity also resulted in increased edge enhancement (N-Gen * 3D$_{variation}$, $\beta = 0.027$, $t_{36210} = 3.283.10$, $p = 0.001$), though unexpectedly this observed increase was greater for mixed lighting treatments for 3D variable habitats (N-Gen * MIXED * 3D$_{variation}$, $\beta = 0.077$, $t_{36210} = 6.59$) and irrespective of the 3D environment (N-Gen * MIXED, $\beta = 0.024$, $t_{36210} = 2.09$, $p = 0.037$). Whether the increased selection for edge enhancement within 3D complex mixed lighting regimes was due to it functioning as a generalist strategy or 3D environments increasing the survival time under direct lighting is difficult to disentangle.

The 'potential' glossiness (change in target luminance value when gloss genes are maximised) was negatively affected by direct lighting and the target-skin's luminance, darker targets inherently appear glossier (see Supplementary Material, S1_File). The relative glossiness of the evolved targets was significantly lower for 3D complex habitats except for those evolved under mixed lighting conditions (Table 3). Targets evolved significantly less-glossy phenotypes over time, but only within 3D complex backgrounds. Notably, diffuse lighting increased selection against glossiness, while mixed lighting did not select against glossiness. Some level of glossiness may help compensate for changes in target and background appearance under variable lighting conditions. Most surviving targets, however, did not appear 'glossy', with extreme glossiness values dying out in the early generations and high internal contrast from patterning masking the specular highlights from gloss

### Discussion

We assessed the effects of three different lighting treatments on the evolution of artificial animal colouration. In doing so, we confirmed that direct lighting on average increases rather than decreases the difficulty of target detection. This was due to direct lighting increasing background complexity and, in doing so, facilitated faster and more reliable camouflage

**Table 2. Linear mixed model output effects of lighting treatment and 3D complexity on the unrendered target-skin's contrast across spatial scales. Positive estimates and t values are shown in blue while negative values are shown in red. Model formula: lmm (patternenergy~Scale*Light*3D+(1|Treatment)+(1|Habitat) +(1|uniquephoto)).**

|  | Predictor | B | SE | DF | T value | P value |
|---|---|---|---|---|---|---|
| **Log Pattern Energy** | Intercept | −0.766 | 0.13 | 76 | −5.84 | <0.001 |
|  | Scale | 0.257 | 0.01 | 9040 | 23.33 | <0.001 |
|  | Mixed | 0.029 | 0.17 | 56 | 0.18 | 0.861 |
|  | Diffuse | −0.085 | 0.17 | 62 | −0.52 | 0.607 |
|  | Mean 3D Var | −0.169 | 0.18 | 75 | −0.96 | 0.341 |
|  | Scale: Mixed | −0.002 | 0.02 | 9040 | −0.15 | 0.885 |
|  | Scale: Diffuse | −0.033 | 0.02 | 9040 | −2.19 | 0.029 |
|  | Scale: 3D | 0.055 | 0.01 | 9040 | 3.73 | <0.001 |
|  | Mixed: 3D | 0.010 | 0.22 | 55 | 0.04 | 0.965 |
|  | Diffuse: 3D | 0.026 | 0.23 | 60 | 0.11 | 0.910 |
|  | Scale: Mixed: 3D | −0.053 | 0.02 | 9040 | −2.50 | 0.012 |
|  | Scale: Diffuse: 3D | −0.111 | 0.02 | 9040 | −5.24 | <0.001 |

**Table 3. Linear mixed model output effects of lighting treatment and background geometry on the evolved glossiness of the targets, relative to the maximum glossiness value they could have. Model formula: lmm (relativeglossiness ~ N-Gen * Light * 3D$_{variation}$ + (1|Treatment) + (1|Habitat)).**

|  | Predictor | B | SE | SD | T value | P value |
|---|---|---|---|---|---|---|
| **Relative Glossiness** | Intercept | 0.028 | 0.09 | 66 | 0.32 | 0.747 |
|  | N-Gen | −0.004 | 0.01 | 36210 | −0.56 | 0.579 |
|  | MIXED | 0.090 | 0.12 | 66 | 0.73 | 0.467 |
|  | DIFFUSE | −0.184 | 0.12 | 66 | −1.49 | 0.142 |
|  | 3D$_{variation}$ | −0.333 | 0.09 | 66 | −3.84 | <0.001 |
|  | N-Gen: MIXED | 0.033 | 0.01 | 36210 | 3.00 | 0.003 |
|  | N-Gen: DIFFUSE | −0.080 | 0.01 | 36210 | −7.10 | <0.001 |
|  | N-Gen: 3D$_{variation}$ | −0.116 | 0.01 | 36210 | −14.71 | <0.001 |
|  | MIXED: 3D$_{variation}$ | 0.391 | 0.12 | 66 | 3.18 | 0.002 |
|  | DIFFUSE: 3D$_{variation}$ | 0.177 | 0.12 | 66 | 1.43 | 0.157 |
|  | N-Gen: Mixed: 3D$_{variation}$ | 0.163 | 0.01 | 36210 | 14.58 | <0.001 |
|  | N-Gen: Diffuse: 3D$_{variation}$ | 0.063 | 0.01 | 36210 | 5.56 | <0.001 |

evolution (fewer failures to improve the fitness proxy, survival time). This provides the first empirical demonstration of Merilaita's [3] oft-cited model for the importance of background complexity in the evolution of camouflage, while also showing shadows to be a powerful source of complexity. In line with our predictions, there were clear differences in the colouration, contrast and orientations of evolved target patterns between lighting treatments. Specifically, camouflage selection under direct lighting leads to more contrasting, darker patterns, countershading, increased edge enhancement and more directional patterns compared to those within diffuse lighting environments (see Fig 4. Bii for examples). These phenotypic features emerged in just 20 generations, demonstrating how alternate camouflage strategies are selected for rapidly under different habitat structures and lighting regimes. The phenotypic differences between evolved populations also corresponded with differences in the predictive power of common camouflage metrics, with edge disruption being more effective against backgrounds with direct lighting.

Our results highlight the critical relationships between lighting and camouflage with habitat geometries. Habitat geometries that produced larger shadows in the presence of direct lighting altered the evolved target patterns, increasing pattern contrast and directionality for both mixed and direct-lighting-only treatments. The increased scale of background spatio-chromatic variation under direct lighting against more 3D complex habitats increased the survival time benefits of camouflage under direct relative to diffuse lighting. This increased disparity between the lighting conditions may have driven the observed increased overlap in phenotype between the mixed and direct-only lighting treatments within more 3D complex habitats. As targets within the mixed lighting treatments were randomly assigned to either a diffuse or direct background, habitats where one lighting condition favours survival over another will have a skewed selection of traits to the lighting with the highest benefit to survival [43,62]. Some components of the mixed lighting evolved targets did differ from the direct within the more 3D complex habitats. For example, pattern contrast at larger spatial scales evolved to be more intermediate between direct and diffuse. Mixed lighting targets also continued to have low levels of glossiness in later generations, despite diffuse lighting and 3D complexity selecting against glossiness for the other treatments. As the luminance of glossy surfaces changes in response to both lighting and geometry, glossiness may serve as a mechanism of maintaining contrast match across different environments. The overall selection against glossiness was unsurprising, with previous experiments showing gloss to negatively impact survival [44] and aversion of shiny materials in favour of matte by the materials [63].

Countershading has long been proposed as an adaptation for camouflage by improving background match and/or by destroying shape-from-shading cues generated by the interaction of the illuminant with geometry [61,64]. We add to the mounting body of evidence that countershading does indeed improve camouflage under direct lighting and that, although mixed lighting selected for reduced countershading, some level of shading gradient persisted, indicating that it can be robust in the face of variable lighting conditions [43]. We also found correlations between countershading and striped patterns, as well as reduced internal contrast under direct lighting for darker targets, irrespective of countershading. Phylogenetic comparisons of ungulate patterns also found a correlation between more intense countershading and stripes [34], suggesting that these two features may interact to disrupt shape.

We also found exceptions to the prediction that the gradient of countershading should perfectly counter self-shading, known as 'optimal countershading' [18]. Against backgrounds where the substrate contained objects that resembled the target, e.g., pebbles and stones, countershading was not selected for, and in some cases, the inverse evolved, despite the intensity of self-shading being higher within these habitats. The presence of objects that match geometric features of the target could negate the need for self-shadow destruction and, by extension, make masquerade conflict with countershading as a camouflage strategy [65]. Indeed, many of the targets observed possessed patterns or colourations that resembled a singular object, particularly stones. Patterns which instead distort the shape of the self-shadow, such as inverse countershading or shadow-like markings, could provide alternative mechanisms for masking shape recognition [7]. However, it is worth noting that our experiment only considers prey viewed from one orientation in plain view, whereas countershading must function against more than one detection angle and in the presence or absence of obstruction. In particular, our targets were viewed from above, rather than from the horizon, as will be the case for many animals such as the ungulates previously studied. Differences in the orientation of shadows within the scene at this orientation may alter which countershading patterns most effectively conceal prey. Field experiments are required to further validate the phenotypic effects found within our experiments, both for countershading and other patterning. Likewise, the effect of body shape and the shape of background distractors on selection for countershading has yet to be explored.

Many animals have directional striped markings which can be orientated to match different background features [66]. Animals which occupy mixed lighting conditions may need to change their orientation and position as clouds pass over, not just for thermoregulation, but to maintain orientation match to the pictorial features of their background [26,64]. The orientation of markings may reflect the typical orientation of the animal relative to the light when basking or navigating its environment, in a similar way to how orientation affects countershading in arboreal animals

[26,67,68]. The increased level of pattern directionality and shifts in pattern orientation observed between our lighting treatments may have been the result of how we orientated the targets. Targets were only orientated with respect to background features for backgrounds with direct lighting. The targets were always aligned parallel to their self-shading gradient so that their countershading patterns would in turn be aligned with them. As a result, targets against direct backgrounds could evolve patterns parallel to the light, matching the orientation of vertical received shadows from vegetation found within 3D complex habitats. Meanwhile, for diffuse backgrounds, the target orientation was random with respect to the background patterns. Other aspects of pattern shape and orientation may have also differed if the targets were allowed to make behavioural decisions on where to position and orient themselves based on the global and/or local statistics of the scene, e.g., moths that align with the patterns of background bark [4,66]. These behavioural decisions are likely to have a large impact on camouflage phenotypes (Kang et al., 2012; Stevens and Ruxton, 2019).

## Conclusion

Overall, our work is the most comprehensive endeavour to link animal camouflage to the physical structure and lighting regimes of habitats. We have been able to show that while there are trends for optimising phenotypes under different conditions, there are also numerous exceptions and alternative strategies that provided effective protection by camouflage. These exceptions potentially act as important drivers of evolutionary diversity, particularly when linked with predator learning and frequency-dependent effects. Several camouflage adaptations, such as masquerade, disruptive colouration, countershading and even colour match, were affected by geometry and lighting environment in ways that were unexpected from previous work [9,18,69]. Patterns that are more contrasting than the background and superficially disruptive can also be pattern matching to the level of background contrast under direct lighting, as shown by the increased pattern contrast of mixed and direct lighting only evolved patterns. We predict that simple ecological and behavioural differences between species in how they interact with their background's geometry under different lighting conditions can, by extension, affect the mechanisms with which they camouflage themselves. Furthermore, any alterations to habitat geometry and lighting from human activity, whether it be shifts in atmospheric condition from climate change or habitat alteration from land-management and urbanisation, could affect the adaptive value of camouflaged phenotypes and the predation risk of vulnerable species. Finally, the methods used here can also be used to explore the effects of and interactions between lighting, geometry and a wide range of alternative drivers of colour evolution, such as phylogenetic constraints, object shape, thermoregulation, signalling and observer condition (visual system and field of view) [31,57,70–72].

## Supporting information

**S1 File. Supporting Information.** Contains additional descriptions for methods and findings not included within the main text.
(PDF)

**S2 File. Supporting Statistics.** Contains the full stats tables for all of the analyses included in the main text.
(PDF)

**S3 File. Supporting Data.** Zip containing all of the data frames and R code necessary to perform our statistical analyses.
(ZIP)

## Acknowledgments

We would like to thank our volunteers for participating in our experiment and our two anonymous volunteers.

## Author contributions

**Conceptualization:** George R.A. Hancock, Innes C. Cuthill, Jolyon Troscianko.

**Data curation:** George R.A. Hancock.

**Formal analysis:** George R.A. Hancock.

**Funding acquisition:** Innes C. Cuthill, Jolyon Troscianko.

**Investigation:** George R.A. Hancock.

**Methodology:** George R.A. Hancock, Innes C. Cuthill, Jolyon Troscianko.

**Resources:** Jolyon Troscianko.

**Software:** George R.A. Hancock, Jolyon Troscianko.

**Supervision:** Innes C. Cuthill, Jolyon Troscianko.

**Validation:** George R.A. Hancock, Jolyon Troscianko.

**Visualization:** George R.A. Hancock.

**Writing – original draft:** George R.A. Hancock.

**Writing – review & editing:** George R.A. Hancock, Innes C. Cuthill, Jolyon Troscianko.

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
