## [Decision Letter · Decision Letter 0]

10 Sep 2025

Dear Dr. Hancock,

We look forward to receiving your revised manuscript.

Kind regards,

Shoko Sugasawa

Academic Editor

PLOS ONE

Journal Requirements:

2. Please note that PLOS ONE has specific guidelines on code sharing for submissions in which author-generated code underpins the findings in the manuscript. In these cases, we expect all author-generated code to be made available without restrictions upon publication of the work.

Please review our guidelines at https://journals.plos.org/plosone/s/materials-and-software-sharing#loc-sharing-code and ensure that your code is shared in a way that follows best practice and facilitates reproducibility and reuse.

“Natural Environment Research Council (NERC) GW4+ NE/S007504/1 funded GRAH in a CASE partnership with the Game and Wildlife Conservancy Trust, UK. JT was funded by a NERC Independent Research Fellowship NE/P018084/1 and ICC by Biotechnology and Biological Research Council grant BB/S00873X/1.”

5. Please note that funding information should not appear in the Acknowledgments section or other areas of your manuscript. We will only publish funding information present in the Funding Statement section of the online submission form. Please remove any funding-related text from the manuscript.

7. We are unable to open your Supporting Information file [S3_Data.zip]. Please kindly revise as necessary and re-upload.

Reviewers' comments:

Reviewer's Responses to Questions

**Comments to the Author**

1. Is the manuscript technically sound, and do the data support the conclusions?

Reviewer #1: Yes

Reviewer #2: Yes

2. Has the statistical analysis been performed appropriately and rigorously?

Reviewer #1: Yes

Reviewer #2: Yes

3. Have the authors made all data underlying the findings in their manuscript fully available?

Reviewer #1: Yes

Reviewer #2: Yes

4. Is the manuscript presented in an intelligible fashion and written in standard English?

Reviewer #1: Yes

Reviewer #2: Yes

Reviewer #1: This study used human participants playing an online object finding game, in combination with genetic algorithms, which allow object pattern and color to evolve, to test the effect of light environment and habitat complexity on the evolution of camouflage. They found that light environment does indeed have considerable effects on the optimal camouflage coloration and pattern, particularly in complex habitats, as shadows add to background complexity. This is a clever and interesting study that provides experimental evidence relevant to multiple theories on the evolution of camouflage.

I do however have a couple of suggestions, as well as some questions for the authors.

The paper was sometimes difficult to follow – although I must stress I do not think this is the fault of the authors, rather it is simply because it contains a lot of information, with multiple predictions and results. I did appreciate Table 1 as it gave a clear overview of the predictions – I wonder however if a table like this could be repeated, including which predictions were proven true/false. Otherwise one has to go back and forth between the table and the results/discussion to get a clear overview of the findings.

Technical language was used in the abstract which could be an issue for the broader readership (this also contrasted with the introduction, which for the most part was much more tailored towards a general audience). I would suggest making the abstract a little less technical.

Could repeat players getting better at the game account for the populations that didn’t show improved fitness over the generations? What information do you have on individual players in the game and are they limited in how many times they can participate? I saw that players could only play one population per generation – but how many players played all generations? This aspect was not discussed at all, and while I know these type of games are now very well-accepted and have been used in a lot of studies I still think it’s worth a couple of lines about your “predator” population.

L198 To change the target from matt to glossy, the RGB values of the glossy targets were pasted with addition (see S1_File). <- I did not understand this sentence.

L284 I also found this hard to follow – perhaps due to a lack of explanation of what the techniques used actually do: target edge disruption was measured using gabrat of the target’s surrounding in the luminance channel, with the default sigma of 3, against its original background (58). <- I think many people (including myself) don’t really know what gabrat does.

L325 is this sentence supposed to trail off…?

L325 – I don’t understand why you included if the slope was negative as a random factor – what is this doing exactly?

Reviewer #2: The authors present a comprehensive computational approach to assess the efficacy of camouflage strategies in conjunction with changes in light and habitat geometry. Given the numerousness and diversity of interactions between these ecological features in the wild, as well as the difficulty in recreating and standardising these interactions in a non-computational experimental set-up, I agree with the claim by the authors that their approach is the first and most optimal solution for addressing this question. The paper therefore presents a very useful foundation for further camouflage investigation and will be of interest to many readers. I have no major concerns with the content, though I believe the clarity of the writing and punctuation needs to be improved at times (outlined in my minor comments). While I understand that the scale of this study and the nature of the question generates a multitude of different statistical comparisons and outputs, I do also fear that the results section may be slightly overwhelming to some readers. I therefore wonder if these results can be summarised with clearer language (i.e., not using model terms) and infographics, as demonstrated in the predictions table (which was very useful for visualisation).

Minor comments

Line 51. Different referencing style for “(Matchette et al., 2018, 2019)”

Line 52. Consider rephrasing to “… maintain effective camouflage, as well as to maximise salience when signalling”

Line 56. Consider rephrasing to “… what should camouflage, which is specialised to either fixed or variable lighting environments, look like”

Line 58. * “On a cloudless day, when the habitat is open, …”

Line 103. * “For example, comparisons of…”

Line 104. There is a big emphasis on countershading within this paper, which is reasonable given its efficacy naturally depends on the light environment. However, the set-up (i.e., viewing targets from a birds-eye-view) is unrepresentative of the most common natural forms of countershading, where the effect is most pronounced when viewed laterally (with the effect representing background matching when viewed from above or below). I agree that, here, a countershading effect may evolve (if direct lighting is present) and that this is informative, but the conditions and context for this emergence here are less naturalistic and I think this needs to be caveated for the reader.

Line 110. * “… enhancement patterns, which pictorially mimic shadows that form at the edges of objects from direct lighting, are harder…”

Line 119. * “…thought to be one factor…”

Line 121. * “change select for background matching…”

Line 126. * “e.g., glossiness (44)”

Line 128. Consider restructuring the sentence introducing the predictions table.

Line 138. I’d move the reference for CamoEvo to directly next to it, then adjust to “which we used to disentangle…”

Line 140. Be consistent with capitalisation for every mention of CamoEvo.

Line 148. * “… by lighting, such as…”

Line 149. This line is a repeat of the earlier statement; consider removing this or the earlier version.

Line 153. * “…directionality, while downwards…”

Line 154. * “… a negative effect; horizontal lines indicate no effect.”

Line 156. The phrase “refer to the axes widely used CIELAB perceptual colour space” needs rewording for increased clarity.

Line 168. Do you have some quantification of how much the intensity of light varied between direct light treatments across habitats?

Line 172. From here on, the use of “Fig” should be adjusted to “Fig.”

Line 174. Did you have some procedure when choosing which habitat patch to photograph (and therefore where the target was placed)?

Line 183. An example would be helpful for “other tall objects”

Line 193. * “glossiness of the background pattern.”

Line 193. Consider rephrasing this sentence e.g., “For full details of this system, see the Supplementary Materials and XXX et al. (XX)”

Line 200. The phrase “were then randomly cropped to create the stimuli” needs more clarity for the reader.

Line 202. Be consistent with language (e.g., “target” or “target button”), or more explicitly introduce the “button”.

Line 208. * “… (DIFFUSE), and then a mixed treatment…”

Line 214. * “.. the previously mentioned stimuli.”

Line 215. What was the source for the existing javascript?

Line 216. * “With this set-up, the goal was to use humans…”

Line 219. * “capturing the targets, one by one, as fast as they could.”

Line 223. * “… displayed in a random order.”

Lines 230-235. I think this section needs a little more clarity. So, if I have this right, ‘response time’ is the time between the start of the trial and the time at which point the cursor breaks the circle, whereas capture time comprises the response time plus the time from circle to the target. It would be good to get a sense of the number of trials that capture time and response time are used respectively. If capture time is used in extreme cases (i.e., delayed or unreasonably fast) as well as for all touch screen interactions (again, an n would be useful), then why not just use capture time throughout (for consistency)? The legend for Figure 1 also states that the selection of targets was based on the time taken to click on the targets, which wouldn’t have been the case for “response time” instances (unless I’ve misunderstood).

Line 234. Be more explicit that the measure is a proxy for survival time (rather than “dubbed”)

Line 237. Wrongly formatted reference.

Line 246. Is “(Gen)” needed here?

Line 302. * “… Dev=1, while lighting treatment was encoded as…”

Lines 302 and beyond. Be consistent with “lighting treatment” – I think it makes it easier to follow.

Line 312. Consider introducing the formula as: “Using lme4 nomenclature, the models were as follows:”. Also, there is no need for the end of this sentence to still be in italics, and can be isolated as a new sentence.

Line 315. * “… geometry (i.e., pattern contrast…”

Line 316. * “… glossiness), we first…”

Line 327. I think the response measures that have been transformed need to be explicitly mentioned (for replicability’s sake)

Line 332. * “… tables are provided in the Supplementary Materials (S2_File).”

Line 336-338. Consider rephrasing this sentence to “To assess whether targets become more camouflaged, we measured the change in survival time (i.e., our fitness proxy) as a function of generation number (Fig. 2).” For the layman (and general readability), I’d also stick to phrases like “generation number” rather than “n-generation” throughout.

Line 337. * “… improved generation number (via a model not using…”

Line 342. Add number of replicates here i.e., “… increase in fitness (n = 3) or decreased in fitness with generation number (n = 12).”

Line 344. Consider rephrasing to “Populations where the generation number lowered survival time were excluded from subsequent analyses, with the exception of the habitat analysis.”

Line 347. Best to choose either variation or complexity (rather than use a forward slash).

Line 360. Sentence ends abruptly after “which”.

Line 365. *than

Line 373. *effect

Line 392-395. Why Ai if only one panel (same applies to later figures)? Bi needs to have slightly more information i.e., including information from the key. The start of the description for Bii is also needs to be made more clear. Change from “pattern match” to “pattern difference” (as this is what is in the figure)

Line 451. * “…for all substrates under both mixed and direct lighting, except gravel under the direct lighting condition”.

Line 453. * high

Line 499. *see

Line 514. * “… of Merilaita’s (3) oft-cited model…”

Line 525. * “… critical relationships between lighting and camouflage with habitat geometry.” (Then delete the rest of that sentence)

Line 545. This paper does not directly test the military aspect, so an additional reference is needed here.

Line 563. * “… and, by extension, make…”

Line 574. * “… has yet…”

Line 577. Is maculation the best word here? This term has been used a few times in the paper, and I think it probably needs to be more clearly defined at its first reference i.e., my image of this word is spots, hence why “directional striped ‘spots’” are hard to visualise

Line 588. * “… turn be aligned with them.”

Line 609. Consider rephrasing to “We predict that simple ecological and behavioural differences…”, mainly because these haven’t been tested.

Supplementary: S1_File

Line 19. * “… used. Sex…”

Line 23. 2/3 and 1/3 better written out as text.

Line 26. First mention of “lvl”, so best to write long form.

Line 60. * “… to evolve, two new…”

Line 64. * “… maculation, the material…”

Line 88. * “… three lighting treatments (direct, mixed, and diffuse), and then….” Also, be consistent the capitalisation of these treatments – the main text includes these in full capitals.

Line 112. The start of this sentence doesn’t seem to make sense.

Line 121. *four

.

Reviewer #1: No

Reviewer #2: No

While revising your submission, please upload your figure files to the Preflight Analysis and Conversion Engine (PACE) digital diagnostic tool, https://pacev2.apexcovantage.com/. PACE helps ensure that figures meet PLOS requirements. To use PACE, you must first register as a user. Registration is free. Then, login and navigate to the UPLOAD tab, where you will find detailed instructions on how to use the tool. If you encounter any issues or have any questions when using PACE, please email PLOS at . PACE helps ensure that figures meet PLOS requirements. To use PACE, you must first register as a user. Registration is free. Then, login and navigate to the UPLOAD tab, where you will find detailed instructions on how to use the tool. If you encounter any issues or have any questions when using PACE, please email PLOS at . PACE helps ensure that figures meet PLOS requirements. To use PACE, you must first register as a user. Registration is free. Then, login and navigate to the UPLOAD tab, where you will find detailed instructions on how to use the tool. If you encounter any issues or have any questions when using PACE, please email PLOS at . PACE helps ensure that figures meet PLOS requirements. To use PACE, you must first register as a user. Registration is free. Then, login and navigate to the UPLOAD tab, where you will find detailed instructions on how to use the tool. If you encounter any issues or have any questions when using PACE, please email PLOS at figures@plos.org. Please note that Supporting Information files do not need this step.. Please note that Supporting Information files do not need this step.

---

## [Author Response · Author response to Decision Letter 1]

20 Feb 2026

Please see attached .doc for formatted version.

Reviewer 1

-------------------

Reviewer #1: This study used human participants playing an online object finding game, in combination with genetic algorithms, which allow object pattern and color to evolve, to test the effect of light environment and habitat complexity on the evolution of camouflage. They found that light environment does indeed have considerable effects on the optimal camouflage coloration and pattern, particularly in complex habitats, as shadows add to background complexity. This is a clever and interesting study that provides experimental evidence relevant to multiple theories on the evolution of camouflage.

I do however have a couple of suggestions, as well as some questions for the authors.

The paper was sometimes difficult to follow – although I must stress I do not think this is the fault of the authors, rather it is simply because it contains a lot of information, with multiple predictions and results. I did appreciate Table 1 as it gave a clear overview of the predictions – I wonder however if a table like this could be repeated, including which predictions were proven true/false. Otherwise one has to go back and forth between the table and the results/discussion to get a clear overview of the findings.

Response: We attempted and trialled several iterations of infographics but we found them to be too large, dense (original infographic did not included mixed lighting) and overly overlapping with the existing figures. We instead opted to make the existing figures a bit clearer.

R1.02. Technical language was used in the abstract which could be an issue for the broader readership (this also contrasted with the introduction, which for the most part was much more tailored towards a general audience). I would suggest making the abstract a little less technical.

Could repeat players getting better at the game account for the populations that didn’t show improved fitness over the generations? What information do you have on individual players in the game and are they limited in how many times they can participate? I saw that players could only play one population per generation – but how many players played all generations? This aspect was not discussed at all, and while I know these type of games are now very well-accepted and have been used in a lot of studies I still think it’s worth a couple of lines about your “predator” population.

Response: Unfortunately, one limitation of running the game online and using anonymous players was that people could indeed play the experiment multiple times. The only information for each play that was gathered was the “survival_time” for each slide. While it is feasible that improvement in individual player performance could have accounted for some of the failure to improve capture time it would not explain the fact that failure to improve was significantly more likely in less complex habitats or the failure to improve in camouflage within these populations as each generation was independently ranked.

We have now amended “(d) Online experiment design:” to reference the fact that players could play multiple times

L198 To change the target from matt to glossy, the RGB values of the glossy targets were pasted with addition (see S1_File). <- I did not understand this sentence.

Response: we have updated the sentence as follows “To change the target from matt to glossy, the image of the glossy target was combined with the patterned target using image addition. To provide a range of glossiness values, the image of the glossy target was multiplied by a genetically controlled alpha value (see S1_File for more details).

L284 I also found this hard to follow – perhaps due to a lack of explanation of what the techniques used actually do: target edge disruption was measured using gabrat of the target’s surrounding in the luminance channel, with the default sigma of 3, against its original background (58). <- I think many people (including myself) don’t really know what gabrat does.

Response: we have now provided a brief description of gabrat. “Gabrat works by quantifying the ratio of “false” edges to “coherent” edges around the object's outline and has been shown to predict search time for camouflaged disruptive prey by humans.”

L325 is this sentence supposed to trail off…?

Repsonse: apologies, yes it was meant to trail off to indicate additional pieces of code e.g. the dataframe. However, we should have closed the bracket. It is now “lmm (phenotypemetric ~ N-Gen * Light * 3D + (1|Habitat) + (1|Population) … ).”

L325 – I don’t understand why you included if the slope was negative as a random factor – what is this doing exactly?

Response: We have updated the sentence as follows. “To account for instances where survival time failed to improve, a binary variable for whether the population had a negative slope for survival time with n-generations was used as a random factor. “

Reviewer 2

-------------------

Reviewer #2: The authors present a comprehensive computational approach to assess the efficacy of camouflage strategies in conjunction with changes in light and habitat geometry. Given the numerousness and diversity of interactions between these ecological features in the wild, as well as the difficulty in recreating and standardising these interactions in a non-computational experimental set-up, I agree with the claim by the authors that their approach is the first and most optimal solution for addressing this question. The paper therefore presents a very useful foundation for further camouflage investigation and will be of interest to many readers. I have no major concerns with the content, though I believe the clarity of the writing and punctuation needs to be improved at times (outlined in my minor comments). While I understand that the scale of this study and the nature of the question generates a multitude of different statistical comparisons and outputs, I do also fear that the results section may be slightly overwhelming to some readers. I therefore wonder if these results can be summarised with clearer language (i.e., not using model terms) and infographics, as demonstrated in the predictions table (which was very useful for visualisation).

Minor comments

Line 51. Different referencing style for “(Matchette et al., 2018, 2019)”

Response: Corrected, it had been accidentally unlinked from the citation manager.

Line 52. Consider rephrasing to “… maintain effective camouflage, as well as to maximise salience when signalling”

Response: rephrased as requested.

Line 56. Consider rephrasing to “… what should camouflage, which is specialised to either fixed or variable lighting environments, look like”

Response: rephrased as requested

Line 58. * “On a cloudless day, when the habitat is open, …”

Response: Modified as requested.

Line 103. * “For example, comparisons of…”

Response: Modified as requested.

Line 104. There is a big emphasis on countershading within this paper, which is reasonable given its efficacy naturally depends on the light environment. However, the set-up (i.e., viewing targets from a birds-eye-view) is unrepresentative of the most common natural forms of countershading, where the effect is most pronounced when viewed laterally (with the effect representing background matching when viewed from above or below). I agree that, here, a countershading effect may evolve (if direct lighting is present) and that this is informative, but the conditions and context for this emergence here are less naturalistic and I think this needs to be caveated for the reader.

Response: We agree that this is a potential caveat of our paper and we reference this in lines 570-572 “It is possible that countershading is still beneficial when viewed from the side where the background appearance differs in visible structure.”. To further highlight this point we have modified the line to emphasise that many camouflaged animals will be viewed from the side “In particular, our targets were viewed from above, rather than from the horizon, as will be the case for many animals such as the ungulates previously studied. Differences in the orientation of shadows within the scene at this orientation may alter which countershading patterns most effectively conceal prey”

Line 110. * “… enhancement patterns, which pictorially mimic shadows that form at the edges of objects from direct lighting, are harder…”

Response: Modified as requested.

Line 119. * “…thought to be one factor…”

Response: Modified as requested.

Line 121. * “change select for background matching…”

Response: We aren’t certain if this recommendation is correct. The sentence has been modified as follows: “But does the stability of information change the intensity of selection for background matching within these channels? “

Line 126. * “e.g., glossiness (44)”

Response: Added missing comma

Line 128. Consider restructuring the sentence introducing the predictions table.

Response: We have revised the table legend accordingly.

Table 1. A summary of the predicted effects of lighting condition, habitat 3D variation and the interaction between direct lighting and 3D variation on the appearance of all animal patterns. Upwards arrows indicate a positive effect on the phenotypic feature, e.g., increased luminance or directionality, while downwards arrows indicate a negative effect; a horizontal line indicates no predicted effect. Larger arrows indicate that increased variation increases the effect size of direct lighting, while smaller arrows indicate a reduction in effect size. For each prediction, the justification and relevant references are provided. CIE L*, a* and b* refer to the mean values of the opponent colour channels used by the CIELAB perceptual colour space (50). These channels are L* (luminance, low = dark, high =light), a* (red-green, low = green, high = red) and b* (blue-yellow, low = blue, high = yellow).

Line 138. I’d move the reference for CamoEvo to directly next to it, then adjust to “which we used to disentangle…”

Response: Adjusted accordingly.

Response: Modified as requested.

Line 140. Be consistent with capitalisation for every mention of CamoEvo.

Response: We’ve ensured all instances now show “CamoEvo”.

Line 148. * “… by lighting, such as…”

Response: Modified as requested.

Line 149. This line is a repeat of the earlier statement; consider removing this or the earlier version.

Response: We are uncertain as to what exactly this refers to.

Line 153. * “…directionality, while downwards…”

Response: Modified as requested.

Line 154. * “… a negative effect; horizontal lines indicate no effect.”

Response: Modified to “; a horizontal line indicates no predicted effect”.

Line 156. The phrase “refer to the axes widely used CIELAB perceptual colour space” needs rewording for increased clarity.

Response: Modified to “CIE L*, a* and b* refer to the mean values of the opponent colour channels used by the CIELAB perceptual colour space (50). These channels are L* (luminance, low = dark, high =light), a* (red-green, low = green, high = red) and b* (blue-yellow, low = blue, high = yellow).”

Line 168. Do you have some quantification of how much the intensity of light varied between direct light treatments across habitats?

Response: Unfortunately, we did not take any additional light intensity measures, e.g. the ambient irradiance of the target location. While it would have been good to include intensity as an additional random effect, the core goal of this study was to look at the influence of the directionality of ambient light rather intensity. Variation in time of day and surrounding 3D structures would have created noise in light intensity, directionality (e.g. lower sun = greater directionality of light), etc. Despite this noise, the overall average effect of directionality and geometry was maintained.

Line 172. From here on, the use of “Fig” should be adjusted to “Fig.”

Response: Modified as requested.

Line 174. Did you have some procedure when choosing which habitat patch to photograph (and therefore where the target was placed)?

Response: Due to weather restrictions (limited availability of <10% cloud coer), restrictions to pre-mapping the sites due to COVID, and the inability to place the photography tent in some locations we weren’t able to pre-map the site areas. Habitat patches were visually identified by possessing similar background features and weren’t allowed to overlap, i.e. the distance from the previous location had to be greater than the dimensions of the photography tent (1.5 m).

Line 183. An example would be helpful for “other tall objects”

Response: Rephrased to “3D variation featured vegetation and other elevated surfaces, e.g., large stones.”

Line 193. * “glossiness of the background pattern.”

Response: Rephrased as requested.

Line 193. Consider rephrasing this sentence e.g., “For full details of this system, see the Supplementary Materials and XXX et al. (XX)”

Response: Rephrased as requested.

Line 200. The phrase “were then randomly cropped to create the stimuli” needs more clarity for the reader.

Response: Sentence has been adjusted as follows “Background images with rendered patterns were then cropped with the target position randomly offset from the centre to ensure the target wasn’t always in the same position (S1_File). Each cropped image was 1904 �1488 px, and the target button had a diameter of 80 px.”

Line 202. Be consistent with language (e.g., “target” or “target button”), or more explicitly introduce the “button”.

Response: Button has now been introduced in (a) Visual Scenes “Within each scene, a 3D-printed 30 mm diameter button-shaped target painted grey with a matt 8% reflectance paint was placed at the centre.”

Line 208. * “… (DIFFUSE), and then a mixed treatment…”

Response: Rephrased as requested.

Line 214. * “.. the previously mentioned stimuli.”

Response: Rephrased as requested.

Line 215. What was the source for the existing javascript?

Response: The sentence has been re-written to make it clear that the game was modified from the following experiment: “An online computer-based search game ( https://CamoEvo.visual-ecology.com/ ) was created by modifying JavaScript used for Troscianko et al., 2020 ‘s crab camouflage experiment. “

Line 216. * “With this set-up, the goal was to use humans…”

Response: Rephrased as requested.

Line 219. * “capturing the targets, one by one, as fast as they could.”

Response: Rephrased as requested.

Line 223. * “… displayed in a random order.”

Response: Rephrased as requested.

Lines 230-235. I think this section needs a little more clarity. So, if I have this right, ‘response time’ is the time between the start of the trial and the time at which point the cursor breaks the circle, whereas capture time comprises the response time plus the time from circle to the target. It would be good to get a sense of the number of trials that capture time and response time are used respectively. If capture time is used in extreme cases (i.e., delayed or unreasonably fast) as well as for all touch screen interactions (again, an n would be useful), then why not just use capture time throughout (for consistency)? The legend for Figure 1 also states that the selection of targets was based on the time taken to click on the targets, which wouldn’t have been the case for “response time” instances (unless I’ve misunderstood).

Response: We have rephrased figure 1 to make it clear that survival time was used. Yes, we agree with the reviewer here and this information we have outputted prior for the CamoEvo toolbox. Unfortunately, the online game was run with an earlier version of CamoEvo before its final release and so did not implement separate outputs for survival, response, and capture time for each generation. Only the ID name of the target and its fitness (survival time). There was supposed to be an Output.txt directory which would save all of this information, in addition to the fitness tables for each population and generation. However, the file froze at generation 3 for unknown reasons.

Line 234. Be more explicit that the measure is a proxy for survival time (rather than “dubbed”)

Response: Rephrased as follows: “This

---

## [Decision Letter · Decision Letter 1]

17 Mar 2026

Shining a light on camouflage evolution: using genetic algorithms to determine the effects of geometry and lighting on optimal camouflage

PONE-D-25-23182R1

Dear Dr. Hancock,

We’re pleased to inform you that your manuscript has been judged scientifically suitable for publication and will be formally accepted for publication once it meets all outstanding technical requirements.

Kind regards,

Shoko Sugasawa

Academic Editor

PLOS One

Additional Editor Comments (optional):

Reviewers' comments:

Reviewer's Responses to Questions

**Comments to the Author**

Reviewer #1: All comments have been addressed

2. Is the manuscript technically sound, and do the data support the conclusions?

Reviewer #1: Yes

3. Has the statistical analysis been performed appropriately and rigorously?

Reviewer #1: Yes

4. Have the authors made all data underlying the findings in their manuscript fully available?

Reviewer #1: (No Response)

5. Is the manuscript presented in an intelligible fashion and written in standard English?

Reviewer #1: Yes

Reviewer #1: The authors have sufficiently addressed all my comments and questions, and I found the resulting manuscript easier to follow. I am therefore happy to recommend this paper for publication.

.

Reviewer #1: No

---

## [Editor Report · Acceptance letter]

PONE-D-25-23182R1

PLOS One

Dear Dr. Hancock,

I'm pleased to inform you that your manuscript has been deemed suitable for publication in PLOS One. Congratulations! Your manuscript is now being handed over to our production team.

Kind regards,

on behalf of

Dr. Shoko Sugasawa

Academic Editor

PLOS One